# Application of Diagnostic Interview for Internet Addiction (DIA) in Clinical Practice for Korean Adolescents

**DOI:** 10.3390/jcm8020202

**Published:** 2019-02-06

**Authors:** Hyera Ryu, Ji Yoon Lee, A Ruem Choi, Sun Ju Chung, Minkyung Park, Soo-Young Bhang, Jun-Gun Kwon, Yong-Sil Kweon, Jung-Seok Choi

**Affiliations:** 1Department of Psychiatry, SMG-SNU Boramae Medical Center, Seoul 07061, Korea; hyera.ryu12@gmail.com (H.R.); idiyuni91@gmail.com (J.Y.L.); choiar90@gmail.com (A.R.C.); sunjujung1991@gmail.com (S.J.C.); reneedrv@gmail.com (M.P.); 2Department of Psychiatry, Eulji General Hospital, Seoul 01830, Korea; dresme@daum.net; 3I Will Center, Seoul Metropolitan Boramae Youth Center, Seoul 07062, Korea; jun@boramyc.or.kr; 4Department of Psychiatry, Uijeongbu St. Mary’s Hospital, The Catholic University of Korea College of Medicine, Gyeonggi 11765, Korea; 5Department of Psychiatry and Behavioral Science, Seoul National University College of Medicine, Seoul 03080, Korea

**Keywords:** internet gaming disorder, semi-structured diagnostic interview, psychometric properties, adolescents

## Abstract

The increased prevalence of Internet Gaming Disorder (IGD) and the inclusion of IGD in DSM-5 and ICD-11 emphasizes the importance of measuring and describing the IGD symptoms. We examined the psychometric properties of the Diagnostic Interview for Internet Addiction (DIA), a semi-structured diagnostic interview tool for IGD, and verified the application of DIA in clinical practice for Korean adolescents. The DIA is conducted in a manner that interviews both adolescents and their caregivers, and each item has a standardized representative question and various examples. It consists of 10 items based on the DSM-5 IGD diagnostic criteria, which is cognitive salience, withdrawal, tolerance, difficulty in regulating use, loss of interest in other activities, persistent use despite negative results, deception regarding Internet/games/SNS use, use of Internet/games/SNS to avoid negative feelings, interference with role performance, and craving. The study included 103 adolescents divided into three subgroups (mild risk, moderate risk, and addicted group) based on the total score of DIA. Demographic and clinical characteristics were compared among the DIA subgroups using the chi-square test and analysis of variance (ANOVA), and correlation analysis was used to examine the associations of IGD symptoms with clinical variables (e.g., impulsivity, aggression, depression, anxiety, self-esteem). The DIA total score was significantly correlated with Internet and smartphone addiction, depression, state anxiety, self-esteem, impulsivity, aggression, and stress. Furthermore, the moderate risk and addicted group showed significantly higher levels of Internet and smartphone addiction, anxiety, depression, impulsivity, aggression, stress, and lower self-esteem compared with the mild risk group. The Junior Temperament and Character Inventory (JTCI), which measures temperament and character traits, revealed that the mild risk group had higher levels of persistence and self-directedness than did the addicted group. Our findings confirmed the psychometric properties of DIA and the application of the DIA classifications in Korean adolescents.

## 1. Introduction

The prevalence of Internet addiction has steadily increased, from 10.4% in 2011 to 12.5% in 2014 in Korea [1], and the prevalence of Internet gaming disorder (IGD) is about 6% in Korean adolescents [2]. The American Psychiatric Association (APA) included IGD as a condition worthy of future study in Section III of the 5th edition of the Diagnostic and Statistical Manual of Mental Disorders (DSM-5), and the draft of 11th revision of the International Classification of Diseases (ICD-11), released in 2018, included the definition of gaming disorder (GD) [3,4]. The growing prevalence of IGD and its recognition as a possible behavioral addiction has increased the importance of describing and measuring the symptoms and severity of the condition. In this context, several researchers have noted that unifying the terminology and developing measurement tools based on the DSM-5 IGD diagnostic criteria are necessary to integrate concepts related to IGD [5,6,7,8].

As part of this effort, two self-report questionnaires, the Internet Gaming Disorder Test (IGD-20) [9] and the Internet Gaming Disorder Scale-Short Form (IGDS9-SF) [10], were developed based on the DSM-5 IGD diagnostic criteria. Although self-report questionnaires are cost-effective and easy to administer, the tool has some limitations. Jeong et al. (2018) recently found a discrepancy between self-report data and the clinical diagnosis of IGD in adolescents due to underreporting as a result of social desirability effects, which is a limitation of self-report questionnaires [11]. Therefore, semi-structured diagnostic interviews tools are needed to measure IGD symptoms more accurately in adolescents.

Several structured and the semi-structured interviews have been developed to assess IGD symptoms. The checklist for the Assessment of Internet and Computer Game Addiction (AICA-C) is a semi-structured interview that assesses six criteria (craving, tolerance, withdrawal, loss of control, preoccupation, and negative consequences) [12]. The Structured Clinical Interview for Internet Gaming Disorder (SCI-IGD) is a 12-item structured interview based on DSM-5 criteria and IGD-related clinical experience [13]. However, these tools were developed for middle school students and are only interviewed for adolescents without including their caregivers. The Diagnostic Interview for Internet Addiction (DIA) was developed to evaluate internet, games, and SNS addiction according to the DSM-5 diagnostic criteria, both children/adolescents and their caregivers are interviewed. It consists of 10 items, as ‘craving’ was added to the 9 DSM-5 criteria for IGD (cognitive salience, withdrawal, tolerance, difficulty in regulating use, loss of interest in other activities, persistent use despite negative results, deception regarding Internet/games/social network site (SNS) use, the use of Internet/games/SNS to avoid negative feelings, and interference with role performance). Each item is rated on a 4-point scale (0: No information; 1: No symptoms; 2: Subthreshold; and 3: Threshold level), and the number of items with a score of 3 (threshold) is calculated as the DIA total score. Previous studies have classified the severity of IGD as mild risk, moderate risk, plain addicted, and severe addicted based on the total number of IGD criteria met [14]. Thus, we adopted the previous categories for the DIA subgroups and verified the application of the DIA classifications in clinical practice by comparing IGD-related psychiatric symptoms among the subgroups.

Several studies have shown that IGD is related to internalizing (e.g., depression, anxiety, stress, and self-esteem) and externalizing (e.g., impulsivity and aggression) problems in adolescents [15,16,17]. Depression is the most common symptom associated with IGD in all age groups, and several studies have found that individuals with IGD experienced more severe depression and anxiety symptoms than did non-addicted individuals [17,18,19,20,21]. A longitudinal study found that depression and anxiety were negative outcomes of Internet gaming [22]. Lam et al. (2009) found that stress-related variables such as family dissatisfaction and recent stressful events were associated with IGD symptoms in adolescents [23]. Moreover, low self-esteem, impulsivity, and aggression are risk factors for IGD in adolescents [22,24,25]. Several studies have found positive correlations between IGD symptoms and personality traits (e.g., sensation seeking, neuroticism, high impulsivity, and high aggressiveness) and negative correlations with extraversion, responsibility, reward dependence, complacency, and self-directedness [25,26,27,28,29]. Thus, we examined the relationship between DIA total scores and internalizing/externalizing problems in adolescents, and compared the various characteristics among the DIA severity subgroups.

We examined the psychometric properties of the DIA in Korean adolescents and verified the application of the DIA classification in clinical practice by comparing the clinical characteristics among subgroups.

## 2. Materials and Methods

### 2.1. Participants and Procedure

We screened children and adolescents (aged 7–18 years) who used excessively internet games and/or smartphones in Clinic I-CURE Centers. All participants were screened using questionnaires pertaining to Internet and smartphone addiction (e.g., the Korean Scale for Internet Addiction for adolescents (K-scale); the Korean Smartphone Addiction Scale (S-scale); and the Internet Addiction Proneness Scale for Adolescents (O_A), which is completed by caregivers). Subjects who scored above the cutoff for addiction on at least one screening questionnaire were enrolled in the study. We enrolled 166 participants between August 2015 and December 2017. Of these, 47 children were excluded because the sample size of children was small and there were differences between the questionnaires in children and adolescents (e.g., BDI versus CDI, etc.), and 16 adolescents with intelligence quotient scores <80 or missing data were excluded from the study. The final analysis included 103 subjects between the ages of 13 and 18 years. In this study, the interviewers consisted of a master’s-level of clinical psychology and every interviewer was trained by addiction and/or child-adolescents psychiatry specialists. A flow chart of the study is shown in Figure 1.

### 2.2. Measurements

#### 2.2.1. Diagnostic interview for Internet Addiction (DIA)

The DIA is a semi-structured diagnostic interview tool consisting of 10 items based on the DSM-5 Section III IGD diagnostic criteria; it is used to assess Internet, games, and SNS addiction symptoms (i.e., cognitive salience, withdrawal, tolerance, difficulty in regulating use, loss of interest in other activities, persistent use despite negative results, deception regarding Internet/gaming/SNS use, use of Internet/gaming/SNS to avoid negative feelings, interference with role performance, and craving). In this study, the internal consistency coefficient of DIA was 0.72.

DIA interviews take about 10–20 min for each subject and their caregivers. Each item has a standardized representative question and various examples, so clinicians can more easily evaluate the score. For example, to assess ‘regulation difficulty in regulating use’, there is a representative question: “Do you feel you should reduce the Internet/Games/SNS, but you can’t reduce the time you spend doing Internet/Games/SNS?” Detailed examples are provided as follows: “I often do more Internet/Games/SNS than I originally thought”, “I often don’t do other activities I was planning because of the Internet/Games/SNS”, “I try to stop the Internet/games/SNS but it is difficult to break”. Table 1 shows the standardized interview script for each item. After interviewing the subjects and caregivers, the clinician calculated the total score to determine whether subjects were addicted to the Internet, games, and/or SNS. Each item was rated on a 4-point scale (0: No information; 1: No symptoms; 2: Subthreshold; 3: Threshold level), and the number of items with a score of 3 was calculated as the total DIA score (range, 0–10). 

#### 2.2.2. Internet and Smartphone Addiction Scales

The Korean Scale for Internet Addiction for adolescents (K-scale), Smartphone Addiction Scale-short form version (SAS-SV), Smartphone Addiction scale (S-scale), Young’s Internet Addiction Test (YIAT), and the Internet Addiction Proneness scale for adolescents (O_A) were used to measure Internet and smartphone addiction symptoms. The K-scale, developed by the National Information Society Agency [30], is a 40-item questionnaire with scores on each item ranging from 1 (not at all) to 4 (always). The SAS-SV is a 10-item scale in which each item is rated on a 6-point Likert scale. Scores above the cutoff values of 31 for males and 33 for females indicate high-risk use [31]. The S-scale is a 15-item questionnaire that measures the level of smartphone addiction on a 4-point Likert scale [32]. The YIAT, developed by Young (1998) [33] and validated in Korean by Kim et al. (2003) [34], consists of 20 items, with higher scores indicating more severe Internet addiction. The O_A consists of 15 items [32]. The internal consistency coefficient of all scales was higher than 0.91 in our study.

#### 2.2.3. Clinical Measurements

We examined internalizing and externalizing problems associated with Internet and smartphone addiction using questionnaires to assess depression, anxiety, self-esteem, impulsivity, aggression, stress, temperament, and personality traits. 

The Beck Depression Inventory-II (BDI-II), developed by Beck et al. [35], is a 21-item questionnaire that measures the severity of depression, with higher scores reflecting more severe symptoms. The BDI-II was validated for Korean adolescents by Lee et al. [36], and the Cronbach’s alpha was 0.56 in this study. The State–Trait Anxiety Inventory-X1 (STAI-X1) is a 20-item tool that measures state anxiety [37]. The Cronbach’s alpha was 0.98 in our study. The Rosenberg Self-Esteem Scale (RSES) [38], which is also translated in Korean, measures perceived self-esteem and self-acceptance, with higher scores reflecting high self-esteem. The Barratt Impulsiveness Scale-II (BIS-II), which consists of 23 items and three subscales (cognitive, motor, and non-planning impulsivity), was developed by Barratt and White [39] and translated into Korean by Lee [40]. The Cronbach’s alpha was 0.99. The Korean version of the Aggression Questionnaire (AQ) consists of 27 items, as two of the original 29 items were excluded [41,42]. The AQ measures physical and verbal aggression, anger, and hostility. The Cronbach’s alpha was 0.98. The Daily Hassles Questionnaire (DHQ) measures stress related to parents, family environment, friends, school, teachers, and school life. It was developed by Rowlison and Felner [43] and modified and validated in Korean adolescents by Han and Yoo [44]. Finally, we used the Junior Temperament and Character Inventory (JTCI) to assess four temperaments (novelty seeking, harm avoidance, reward dependence, and persistence) and three character traits (self-directedness, cooperativeness, and self-transcendence) in adolescents. The Korean version of the JTCI consists of 82 items, each with ‘yes’ or ‘no’ response options. 

### 2.3. Statistical Analysis

The chi-square test and analysis of variance (ANOVA) including post-hoc test (Bonferronni method) were used to compare demographic and clinical characteristics among the DIA subgroups and to assess the application of the DIA subgroup classifications in clinical practice. The psychometric properties of the DIA were examined using internal consistency analysis and Pearson’s correlation analysis to assess the association of IGD symptoms (DIA) with Internet and smartphone addiction (K, SAS-SV, S, YIAT, O_A), impulsivity (BIS-11), aggression (AQ), depression (BDI), anxiety (STAI), self-esteem (RSES), stress (DHQ), and temperament/character traits (JTCI). All statistical tests were performed using SPSS software version 21.0 (SPSS, Inc., Chicago, IL, USA).

### 2.4. Ethical Approval

All subjects and their caregivers gave their informed consent for inclusion before they participated in the study. The study was conducted in accordance with the Declaration of Helsinki, and the protocol was approved by the Institutional Review Board (IRB) for human subjects of Uijeonbu St. Mary’s Hospital (UC150NMI0072), Eulji University Eulji Hospital (EMCS2015-05-020-001) and SMG-SNU Boramae Medical Center (16-2016-4).

## 3. Results

### 3.1. Demographic Characteristics

The study included 103 adolescents (mean age, 14.35 ± 1.43 years; 70.9% males) and their caregivers (mean age 46.57 ± 7.69 years; 7.6% males). The Korean Wechsler Intelligence Scale for Children-fourth edition (K-WAIS-IV, 100.03 ± 13.52) was administered to 64 subjects under 16 years of age, and the Korean version of the Wechsler Adult Intelligence Scale, 4th edition (K-WISC-IV, 107.13 ± 12.51) was administered to the remaining subjects. Subjects with DIA total scores of 0–2 were categorized as mild risk, those with scores of 3–4 were moderate risk, and those with scores of 5–10 were classified as addicted. We found no significant differences in sex, age, caregiver’s age, or IQ among the three subgroups (mild risk, moderate risk and addicted). 

### 3.2. Psychometric Properties of the Diagnostic Interview for Internet Addiction

The convergent validity of the DIA was assessed by comparing total DIA scores with scores on other scales measuring Internet and smartphone addiction. The correlation coefficients between the DIA and the other scales were as follows: K-scale, 0.426 (*p* < 0.01); SAS-SV, 0.205 (*p* < 0.05); S-scale, 0.234 (*p* < 0.05); Y-IAT, 0.390 (*p* < 0.01); and O_A, 0.343 (*p* < 0.01; Table 2). 

Additionally, the correlation analysis performed to examine construct validity revealed significant relationships of the DIA score with the BDI-II (*r* = 0.285, *p* < 0.01), STAI_X1 (*r* = 0.294, *p* < 0.01), RSES (*r* = −0.312, *p* < 0.01), BIS-II (*r* = 0.278, *p* < 0.01), AQ (*r* = 0.256, *p* < 0.05), and DHQ (*r* = 0.283, *p* < 0.01) scores (Table 3). These findings suggest that the presence of several IGD symptoms is associated with higher levels of depression, anxiety, impulsivity, aggression, stress, and low self-esteem.

### 3.3. Comparison of Clinical Variables among the Diagnostic Interview for Internet Addiction Subgroups

The participants were divided into three subgroups (mild risk, moderate risk, addicted) according to their DIA total score. Comparisons of the clinical variables (internalizing/externalizing problems and temperament and character traits) among the DIA subgroups are shown in Table 4. We found significant differences in Internet and smartphone addiction, depression, anxiety, self-esteem, impulsivity, aggression, and stress among the subgroups on all of the scales except the SAS-SV. The moderate risk and addicted group had significantly higher levels of Internet and smartphone addiction, anxiety, depression, impulsivity, aggression, and stress and lower self-esteem compared with the mild risk group. Moreover, scores on the JTCI, which measures temperament and character traits, revealed that the mild risk group had significantly higher levels of persistence, self-directedness, and cooperativeness than the addicted group did.

## 4. Discussion

In this study, we examined the psychometric properties of the DIA and verified the application of the DIA severity classifications in Korean adolescents. The main findings and implications of our study are as follows.

First, in this study, the reliability and validity of DIA were verified. The Cronbach alpha of DIA was 0.72, which means that the DIA meets the internal consistency reliability. Also, we found positive associations between the total DIA scores and scores on other Internet and smartphone addiction scales in Korean adolescents. These findings support the convergent validity of the DIA and are consistent with those of previous studies investigating the relationship between IGD symptoms and Internet/smartphone addiction scales [9,10,45]. In particular, Cho et al. (2014) found that the self-diagnostic Internet addiction scale based on the DSM-5 criteria for IGD was positively correlated with the K-scale, which measures Internet addiction, in Korean middle school students (ages 13 and 14 years) [45]. Kim et al. (2016) reported similar findings in adults [46].

We assessed the construct validity of the DIA by examining the relationship of DIA results with depression, anxiety, stress, impulsivity, aggression, and self-esteem. We found that the subjects who reported more severe IGD symptoms as measured by the DIA had higher levels of depression, anxiety, stress, impulsivity, and aggression. Several studies have shown a relationship between IGD symptoms and psychiatric comorbidity; in particular, one review paper found 92% studies described a significant correlation between IGD symptoms and anxiety and 89% studies with depression [47]. Depression is strongly associated with IGD symptoms, and is the most significant factor associated with the development of online gaming addiction in adolescents [18,48,49,50]. Gentile et al. (2011) reported that depression and anxiety levels increased in adolescents who became and remained problematic gamers, and Mehroof et al. (2010) found that state anxiety was significantly associated with online gaming addiction [16,22]. Moreover, Internet addiction may contribute to stress. Individuals who experience anxiety and stress have difficulty communicating and interacting with others in healthy and positive ways [51,52]. Furthermore, several studies have shown that impulsivity and aggression are related to Internet addiction in adolescents and increase vulnerability to IGD [22,53,54,55]. We found a negative correlation between the total DIA score and self-esteem, suggesting that more severe IGD symptoms are associated with lower self-esteem. Similarly, several studies have found a significant relationship between low self-esteem and IGD symptoms such that low self-esteem has been shown to predict the emergence of Internet addiction [25]. Therefore, in this study, the psychometric properties of the DIA were verified by examining the reliability and validity of DIA.

In the context of the application of the DIA classifications in clinical practice for Korean adolescents, we found that subjects in the moderate risk and addicted group had lower self-esteem and significantly higher levels of Internet and smartphone addiction, anxiety, depression, impulsivity, aggression, and stress than did the mild risk group. These findings are similar to those of previous studies mentioned above [16,18,22,47,48,49,50,51,52,53,54,55], suggesting that early intervention is required when the total DIA score is 3 or above. This is because adolescents who were included in the moderate risk group reported internalizing and externalizing problems similar to the addicted group in DIA. 

Previous studies have found associations between Internet addiction and personality traits (e.g., sensation seeking, reward dependence, and self-directedness) [26,27,28]. Similarly, we found that persistence, self-directedness, and cooperativeness as measured by the JTCI were significantly higher in the mild risk group than in the addicted group. In previous studies of the relationship between personality traits and IGD symptoms, Montag et al. (2011) found that IGD scores were negatively correlated with self-directedness and cooperativeness, and Jimenez-Murcia et al. (2014) found that low self-directedness predicted high IGD scores in video game users [56,57]. These findings indicate that subjects who are in the mild risk group, as classified by the DIA, tend to persist in behavior without sustained reinforcement and are able to control their behavior, unlike those who are in the internet/games/SNS addicted group. Moreover, several studies have shown that previous reports of a relationship between IGD symptoms and novelty seeking, a personality feature linked to impulsivity [58], were inconsistent [16,59,60]. It may be that IGD symptoms are unrelated to novelty seeking, or they may be associated with both high and low novelty seeking, leading to the appearance of no relationship. We found no significant differences in novelty seeking among the DIA subgroups in the present study. These findings suggest that impulsive people may not derive fulfillment or enjoyment from games such as the massively multiplayer online role-playing games (MMORPGs), depending on their psychological profile. Indeed, Billieux et al. (2015) found that the behaviors of online gamers were heterogeneous; individuals with IGD had various psychological profiles, including with regard to novelty seeking, and IGD symptoms differed according to these profiles [61].

Our study has several limitations. First, our subjects were screened using an Internet/smartphone addiction scale. Thus, it may be difficult to generalize our findings to non-clinical populations because our study did not include a control group. In addition, the proportion of the addicted group was about 80% or more, because the internet/games/SNS high-risk users were recruited in this study. Therefore, additional analysis (e.g., factor analysis, ROC curve etc.) should be made to propose a cut-off score of DIA or to make it more useful in a clinical setting. Second, previous studies have shown that the severity of IGD differs among game genres [62,63]. Lope-Fernandez et al. (2014) reported that subjects who played MMORPGs spent more time playing and had significantly higher scores on the IGD-20 [64]. We did not investigate differences in game genres; further research is needed to examine the DIA subgroups according to the game media type (e.g., internet, games, SNS etc.), various game genres, gaming patterns, and causes of use. 

Despite these limitations, we examined the psychometric properties of the DIA, a semi-structured tool, and investigated the relationship between DIA total score and a wide range of clinical characteristics. In addition, although the DSM-5 diagnostic criteria for IGD require the presence of five or more symptoms, our findings suggest that early intervention and continuous observation are advisable for individuals with three or more symptoms according to the DIA. 

## Figures and Tables

**Figure 1 jcm-08-00202-f001:**
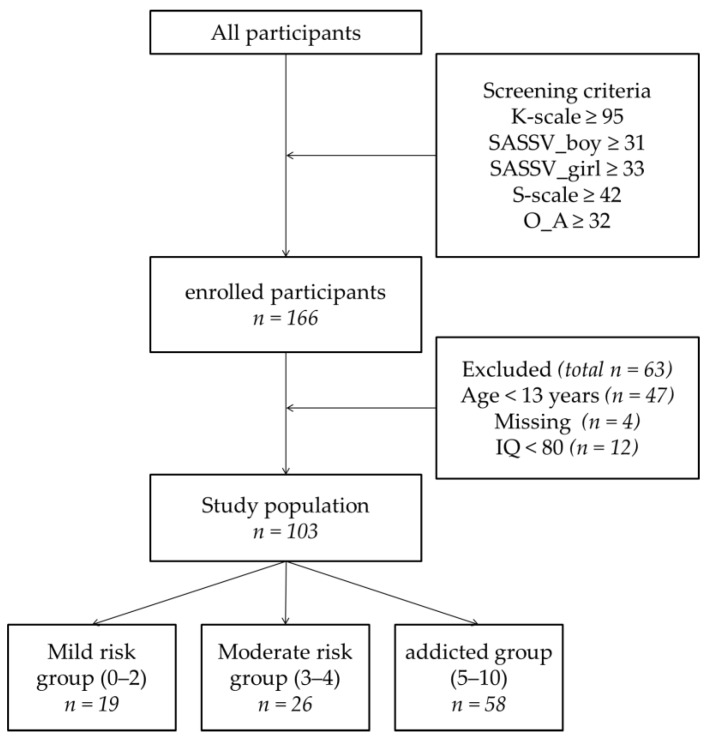
Study flow chart. NOTE: The screening cut-off values shown are those for adolescents; children were excluded from this analysis, and their scores are not reported here. The study participants were divided into subgroups according to their total DIA score. K-scale = Korean Scale for Internet Addiction for adolescents; SAS-SV = Smartphone Addiction Scale-short form version; S-scale = Korean Smartphone Addiction scale; O_A = Internet Addiction Proneness Scale for Adolescents checked by caregivers.

**Table 1 jcm-08-00202-t001:** Examples of standardized interview script in Diagnostic Interview for Internet Addiction.

Item	Standardized Representative Questions
1. cognitive salience	“Even if you do not have Internet/Games/SNS, do you spend a lot of time thinking about Internet/Games/SNS or planning what to do next?”
2. withdrawal	“Do you experience restlessness, irritability, depression, anxiety, sadness etc. when you reduce, stop, or not allowed Internet/Games/SNS?”
3. tolerance	"Do you want to spend more Internet/Gaming/SNS time, find more interesting things, or use better equipment such as cell phones, computers to make you feel as fun as before?”
4. difficulty in regulating use	“Do you feel you should reduce the Internet/Games/SNS, but you can’t reduce the time you spend doing Internet/Games/SNS?”
5. loss of interest in other activities	“Because of the Internet/Games/SNS, would you be less interested in participating in other leisure activities such as hobbies or meet friends?”
6. persistent use despite negative feelings	“Despite negative consequences, such as lack of sleep time, late school or work, spend too much money, debate with other people, or neglect important things, do you continue the Internet/Games/SNS?”
7. deception regarding Internet/gaming/SNS use	“Do you lie or hide how much time you spend on the Internet/Games/SNS for your family or friends?”
8. use of Internet/gaming/SNS to avoid negative feelings	“Do you use Internet games to avoid/relieve negative feelings?" “Do you use the game to forget unpleasant moods (e.g., helplessness, depression, guilty, anxiety, etc.)?”
9. interference with role performance	“Have you ever been troubled or fallen out by the use of Internet /Games/SNS in your important interpersonal, career, and academic settings?”
10. craving	“Do you have a strong desire to do activities such as internet/ Games/SNS?” “If you want to play internet/Games/SNS, is it hard to tolerate?”

**Table 2 jcm-08-00202-t002:** Correlation analysis between Diagnostic Interview for Internet Addiction and Internet and Smartphone related scale.

	DIA	K	SAS-SV	S	YIAT	O_A
**DIA**	1					
**K**	0.426 **	1				
**SAS-SV**	0.205 *	0.638 **	1			
**S**	0.234 *	0.733 **	0.885 **	1		
**YIAT**	0.390 **	0.845 **	0.706 **	0.744 **	1	
**O_A**	0.343 **	0.084	−0.223 *	−0.146	0.047	1

* *p* < 0.05, ** *p* < 0.01. Note. *n* = 103. DIA = Diagnostic Interview for Internet Addiction; K = Korean Scale for Internet Addiction; SAS-SV = Smartphone Addiction Scale-Short form Version; S = Korean Smartphone Addiction scale; YIAT = Young’s Internet Addiction Test; O_A = Internet Addiction Proneness Scale for Adolescents.

**Table 3 jcm-08-00202-t003:** Correlation analysis between the Diagnostic Interview for Internet Addiction and Clinical symptoms.

	DIA	BDI-II	STAI_X1	RSES	BIS-II	AQ	DHQ
**DIA**	1						
**BDI-II**	0.285 **	1					
**STAI_X1**	0.294 **	0.758 **	1				
**RSES**	−0.312 **	−0.708 **	−0.739 **	1			
**BIS-II**	0.278 **	0.390 **	0.422 **	−0.540 **	1		
**AQ**	0.256 *	0.429 **	0.387 **	−0.369 **	0.380 **	1	
**DHQ**	0.283 **	0.538 **	0.595 **	−0.465 **	0.287 **	0.506 **	1

* *p* < 0.05, ** *p* < 0.01. Note. *n* = 103. DIA = Diagnostic Interview for Internet Addiction; BDI-II = Beck Depression Inventory-II; STAI_X1 = State-Trait Anxiety Inventory X−1; RSES = Rosenberg Self-Esteem Scale; BIS = Barratt Impulsiveness Scale-II; AQ = Aggression Questionnaire; DHQ = Daily Hassles Questionnaire.

**Table 4 jcm-08-00202-t004:** Differences in Internet Gaming Disorder symptoms and Clinical variables between Diagnostic Interview for Internet Addiction subgroups.

	Mild Risk M (SD)	Moderate Risk M (SD)	Addicted M (SD)	Total M (SD)	F (Post Hoc)
**K**	58.15 (13.53)	75.57 (18.73)	78.74 (19.98)	74.14 (20.03)	8.807 ** (1<2.3)
**S**	28.52 (7.45)	35.11 (7.86)	34.46 (9.50)	33.53 (9.01)	3.848 * (1<2.3)
**YIAT**	32.76 (10.50)	49.69 (14.13)	50.71 (16.77)	47.14 (16.52)	9.383 ** (1<2.3)
**O_A**	37.57 (8.00)	40.36 (7.40)	43.22 (6.31)	41.47 (7.20)	5.185 * (1<3)
**BDI-II**	6.47 (5.79)	15.69 (11.59)	16.18 (12.90)	14.21 (12.04)	5.267 * (1<2.3)
**STAI_X1**	36.47 (9.60)	44.32 (11.71)	45.83 (12.19)	43.63 (12.05)	4.613 * (1<3)
**RSES**	30.52 (4.38)	26.88 (4.98)	26.31 (6.33)	27.27 (5.85)	3.937 * (1>3)
**BIS**	48.31 (6.61)	57.52 (10.32)	56.81 (9.49)	55.34 (9.78)	6.864 * (1<2.3)
**AQ**	50.36 (12.51)	62.56 (12.62)	62.29 (17.83)	60.05 (16.28)	4.468 * (1<2.3)
**DHQ**	55.84 (15.55)	70.76 (15.70)	70.77(17.43)	67.87 (17.52)	6.145 * (1<2.3)
**JTCI_P**	54.55 (6.42)	47.68 (7.58)	45.22 (8.71)	47.86 (8.66)	8.743 ** (1>2.3)
**JTCI_SD**	55.88 (10.80)	48.68 (10.39)	46.15 (10.81)	48.89 (11.21)	5.292 * (1>3)
**JTCI_C**	56.88 (11.70)	51.20 (8.84)	48.81 (10.10)	51.17 (10.47)	4.065 * (1>3)

* *p* < 0.05, ** *p* < 0.01. Note. No. of mild risk group = 19; No. of moderate risk group = 26; No. of addicted group = 58. Bonferroni post-hoc test results are reported. DIA = Diagnostic Interview for Internet Addiction; K = Korean Scale for Internet Addiction; S = Korean Smartphone Addiction scale; YIAT = Young’s Internet Addiction Test; O_A = Internet Addiction Proneness Scale for Adolescents; BDI-II = Beck Depression Inventory-II; STAI_X1 = State-Trait Anxiety Inventory X−1; RSES = Rosenberg Self-Esteem Scale; BIS = Barratt Impulsiveness Scale-II; AQ = Aggression Questionnaire; DHQ = Daily Hassles Questionnaire; JTCI_P = Junior Temperament and Character Inventory_Persistence; SD = Self-Directedness; C = Cooperativeness.

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
