# Peer review of "Application of Diagnostic Interview for Internet Addiction (DIA) in Clinical Practice for Korean Adolescents"

_jcm, 2019, doi:10.3390/jcm8020202_

Round 1
Reviewer 1 Report
The paper is an interesting example of sharing experiences from the clinical practice. The game addiction issue is also essential from the public health point of view. However, the paper cannot be recommended as standard of research to be applied in other centres. Description of the interview procedure of the as well as patients’ qualification to four groups is not precise and it is not based on clear objective criteria.
My main comments are below:
·There is lack of justification of using the „clinical usefulness” in the title. The research does not give unambiguous evidences. I suggest a simple title: „Application of Diagnostic Interview for Internet Addiction (DIA) in clinical practice”.
· Additionally, „clinical usefulness” misleads the reader. The detailed statistical analysis of diagnosis test can be expected (for example definition of thresholds using ROC curve).
· The full psychometric analysis has not been done. Only some elements of validity occur. The factor structure of and internal consistency should be checked.
· The procedure of the interview has not been described as well as any attempts to standardize it. How long does the interview last? How many clinicians were involved? Was the standardized interview script developed? We only know the scope of 10 issues (omitted in the abstract).
· Adding the script of the interview would improve the quality of this paper.
· Scoring 10 issues is very subjective and can be linked with competences and experience of the expert. Recording interviews and their assessment using independent experts would be a great solution. The inter-rater agreement could be given then. During qualitative studies the interviews are often recorded. Here, it is mentioned that the assessment has been performed after the interview (when?), what can generate errors in case of 10 different criteria.
· I understand that each person was scored in 1-30 scale. However, the 1-10 scale identifying persons who got the worst score in particular issue was finally used. It is a huge wasting of clinical information. The question arises about the total score, its mean and range? An interesting additions of the article can be the table comparing 4 groups according to general score 0-30 and in each of 10 issue. Maybe in this way some determinants of addiction symptoms will be found.
· There is a lack of information why 4 groups, not three or even two (one threshold) was accepted. It can be assumed that authors based on distribution and they wanted to received more or less equal groups. The key table with results (no.3) does not confirm the legitimacy of division into so many groups. Mainly 1 group differs from the others. Only according to test K, the differences between group 3 and 4 were demonstrated.
· The frequency of four groups occurs only in diagram.
· The data has not been interpreted properly. The third group „plain addicted” is functioning better or at least on the same level as the second group.
· There is a lack of explicit recommendation what level should be claimed as critical. If it is 3+ serious symptoms, it qualifies 81,6% of all participants. All the more, the clinical usefulness of this method is doubtful and the problem of no ROC curves comes back.
· In methods it is not said that the post hoc analysis has been applied, although according to the table – it has.
· The summary also arose some objections but its content should be fitted to corrected version of the paper.
The paper has many faults and it needs general improvement. In relations to above mentioned comments I suggest: title change, redefining of aims, more precise description of procedure as well as the method of analysis, introducing the criteria of group division or revision of this division. Some of above comments can be taken into account in description of the study limitation.
Author Response
We highly appreciate your critical review that could be helpful to improve our paper. As you commented, we realized that our previous version of the manuscript had shortcomings and we tried to elaborate the manuscript throughout the paper. Detailed contents revised are as below (also marked in the revised manuscript using track-change function in MS Word).
Reviewer #1
The paper is an interesting example of sharing experiences from the clinical practice. The game addiction issue is also essential from the public health point of view. However, the paper cannot be recommended as standard of research to be applied in other centres. Description of the interview procedure of the as well as patients’ qualification to four groups is not precise and it is not based on clear objective criteria.
My main comments are below:
Comment 1: There is lack of justification of using the „clinical usefulness” in the title. The research does not give unambiguous evidences. I suggest a simple title: „Application of Diagnostic Interview for Internet Addiction (DIA) in clinical practice”. Additionally, „clinical usefulness” misleads the reader. The detailed statistical analysis of diagnosis test can be expected (for example definition of thresholds using ROC curve).
>>> Response 1: Thank you for your comments. We modified the title to be more concise so that there is no confusion
→ "Application of Diagnostic Interview for Internet Addiction (DIA) in clinical practice for Korean adolescents” (in the Tittle)
Comment 2: The full psychometric analysis has not been done. Only some elements of validity occur. The factor structure of and internal consistency should be checked.
>>> Response 2: We appreciate your comment. We added the internal consistency coefficient of DIA to address psychometric properties.
→ “The internal consistency coefficient of DIA was 0.72. (in the measurement, page4 , line 154)
→ “ in this study, the reliability and validity of DIA were verified. The Cronbach alpha of DIA was 0.72, which means that the DIA meets the internal consistency reliability.” (in the conclusion, page 9, line 291-292)
Comment 3: The procedure of the interview has not been described as well as any attempts to standardize it. How long does the interview last? How many clinicians were involved? Was the standardized interview script developed? We only know the scope of 10 issues (omitted in the abstract). Adding the script of the interview would improve the quality of this paper.
>>> Response 3: As you commented, we have added specific description of the DIA interview process and interview script (Table 1) in the “procedure” and revised the “abstract” as follows.
→ “The DIA is a semi-structured interview tool that is conducted in a manner that interviews both adolescents and their caregivers. It consists of 10 items based on the DSM-5 IGD diagnostic criteria, which is cognitive salience, withdrawal, tolerance, difficulty in regulating use, loss of interest in other activities, persistent use despite negative results, deception regarding Internet/gaming/SNS use, use of Internet/gaming/SNS to avoid negative feelings, interference with role performance, and craving. Also, each item has a representative question and various examples” (added in the abstract, page 1, line 31-37)
→”In this study, the interviewers consisted of a master’s-level of clinical psychologist and every interviewer was intensively trained by addiction and/or child-adolescents psychiatry specialists (in the procedure, page 3, line 134-136)
→ “DIA interviews take about 10-20 minutes for each subject and their caregivers. Each item has a representative question and various examples, so clinicians can more easily evaluate the score. For example, to assess ‘the difficulty of regulation’, there is a representative question: "Do you feel you should reduce the Internet/Games/SNS, but you can’t reduce the time you spend doing Internet/Games/SNS?" Detailed examples are provided as follow : "I often do more Internet/Games/SNS than I originally thought", "I often don’t do other activities I was planning because of the Internet/ Games/SNS"," I try to quit the Internet/games/SNS but it is difficult to break". ". Table 1 shows the representative questions of each item.” (in the measurements pate 4, line 155-162)
Table 1. Examples of standardized interview script in DIA.
Item | Standardized representative questions |
1. cognitive salience | “Even if you do not have Internet/Games/SNS, do you spend a lot of time thinking about Internet/Games/SNS or planning what to do next?” |
2. withdrawal | “Do you experience restlessness, irritability, depression, anxiety, sadness etc. when you reduce, stop, or not allowed Internet/Games/SNS?” |
3. tolerance | "Do you want to spend more Internet/Gaming/SNS time, find more interesting things, or use better equipment such as cell phones, computers to make you feel as fun as before?” |
4. difficulty in regulating use | "Do you feel you should reduce the Internet/Games/SNS, but you can’t reduce the time you spend doing Internet/Games/SNS?" |
5. loss of interest in other activities | "Because of the Internet/Games/SNS, would you be less interested in participating in other leisure activities such as hobbies or meet friends?" |
6. persistent use despite negative feelings | "Despite negative consequences, such as lack of sleep time, late school or work, spend too much money, debate with other people, or neglect of important things, do you continue the Internet/Games/SNS?" |
7. deception regarding Internet/gaming/SNS use | "Do you lie or hide how much time you spend on the Internet/Games/SNS for your family or friends?" |
8. use of Internet/gaming/SNS to avoid negative feelings | "Do you use Internet games to avoid/relieve negative feelings?" “Do you use the game to forget unpleasant moods (e.g., helplessness, depression, guilty, anxiety, etc.)?” |
9. interference with role performance | "Have you ever been troubled or fallen out by the use of Internet /Games/SNS in your important interpersonal, career, and academic settings?" |
10. craving | “Do you have a strong desire to do activities such as internet/ Games/SNS?” “If you want to play internet/Games/SNS, is it hard to tolerate?” |
Comment 4: Scoring 10 issues is very subjective and can be linked with competences and experience of the expert. Recording interviews and their assessment using independent experts would be a great solution. The inter-rater agreement could be given then. During qualitative studies the interviews are often recorded. Here, it is mentioned that the assessment has been performed after the interview (when?), what can generate errors in case of 10 different criteria.
>>> Response 4: We appreciate your comment. The DIA presented a representative questions and detailed examples of each items to reduce differences between interviewers. The details of the content have been added to the procedure section as mentioned above (comment 3).
Comment 5: I understand that each person was scored in 1-30 scale. However, the 1-10 scale identifying persons who got the worst score in particular issue was finally used. It is a huge wasting of clinical information. The question arises about the total score, its mean and range? An interesting additions of the article can be the table comparing 4 groups according to general score 0-30 and in each of 10 issue. Maybe in this way some determinants of addiction symptoms will be found.
>>> Response 5: In this study, DIA is rated at 0-3 points (0: no information 1: no symptoms 2: subthreshold 3: threshold level), and the DIA is an interview tool to diagnose the IGD, so it would be calculated by adding the items at threshold level. Thus, the DIA total score range from 0 to 10, the mean of DIA total score was 4.75 (SD=2.32) in this study. As you have mentioned, there is a possibility of confusion in DIA’s scoring method for the readers, so we tried to explain the scoring method of DIA in the manuscript as below.
→ After interviewing the subjects and caregivers, the clinician calculated the total score to determine whether subjects were addicted to the Internet, games, and/or SNS. Each item was rated on a 4-point scale (0: no information; 1: no symptoms; 2: subthreshold; 3: threshold level), and the number of items with a score of 3 was calculated as the total DIA score (range, 0–10). (in the measurement, page 4, line 162-166)
Comment 6: There is a lack of information why 4 groups, not three or even two (one threshold) was accepted. It can be assumed that authors based on distribution and they wanted to received more or less equal groups. The key table with results (no.3) does not confirm the legitimacy of division into so many groups. Mainly 1 group differs from the others. Only according to test K, the differences between group 3 and 4 were demonstrated. The frequency of four groups occurs only in diagram.
>>> Response 6: Thank you for your comments. In this study, we divided into 4 groups based on the previous study*, but in the previous research, there was no significant difference between the plain addicted and severe addicted group. In addition, In the case of more than 5 items in the DIA based on the IGD diagnostic criteria, IGD diagnosis is possible. Thus, we modified the 4 subgroups into 3 subgroups by adding plain addicted and severe addicted group. Also, we modified the Table 4 and described only the significant variables in order to present the results more simply.
* Lee, S.-Y.; Lee, H.K.; Jeong, H.; Yim, H.W.; Bhang, S.-Y.; Jo, S.-J.; Baek, K.-Y.; Kim, E.; Kim, M.S.; Choi, J.-S. The hierarchical implications of Internet gaming disorder criteria: which indicate more severe pathology? Psychiatry Investig. 2017, 14, 249-259
Table 4. Differences in IGD symptoms and Clinical variables between DIA subgroups
Mild risk M (SD) | Moderate risk M (SD) | addicted M (SD) | Total M (SD) | F (post hoc) | |
K | 58.15 (13.53) | 75.57 (18.73) | 78.74 (19.98) | 74.14 (20.03) | 8.807** (1<2,3)< span=""> |
S | 28.52 (7.45) | 35.11 (7.86) | 34.46 (9.50) | 33.53 (9.01) | 3.848* (1<2,3)< span=""> |
YIAT | 32.76 (10.50) | 49.69 (14.13) | 50.71 (16.77) | 47.14 (16.52) | 9.383** (1<2,3)< span=""> |
O_A | 37.57 (8.00) | 40.36 (7.40) | 43.22 (6.31) | 41.47 (7.20) | 5.185* (1<3)< span=""> |
BDI-II | 6.47 (5.79) | 15.69 (11.59) | 16.18 (12.90) | 14.21 (12.04) | 5.267* (1<2,3)< span=""> |
STAI_X1 | 36.47 (9.60) | 44.32 (11.71) | 45.83 (12.19) | 43.63 (12.05) | 4.613* (1<3)< span=""> |
RSES | 30.52 (4.38) | 26.88 (4.98) | 26.31 (6.33) | 27.27 (5.85) | 3.937* (1>3) |
BIS | 48.31 (6.61) | 57.52 (10.32) | 56.81 (9.49) | 55.34 (9.78) | 6.864* (1<2,3)< span=""> |
AQ | 50.36 (12.51) | 62.56 (12.62) | 62.29 (17.83) | 60.05 (16.28) | 4.468* (1<2,3)< span=""> |
DHQ | 55.84 (15.55) | 70.76 (15.70) | 70.77(17.43) | 67.87 (17.52) | 6.145* (1<2,3)< span=""> |
JTCI_P | 54.55 (6.42) | 47.68 (7.58) | 45.22 (8.71) | 47.86 (8.66) | 8.743**(1>2,3) |
JTCI_SD | 55.88 (10.80) | 48.68 (10.39) | 46.15 (10.81) | 48.89 (11.21) | 5.292* (1>3) |
JTCI_C | 56.88 (11.70) | 51.20 (8.84) | 48.81 (10.10) | 51.17 (10.47) | 4.065* (1>3) |
* P<.05, ** P<.01. Note. No. of mild risk group = 19; No. of moderate risk group = 26; No. of addicted group = 58. Bonferroni post-hoc test results are reported. DIA = Diagnostic Interview for Internet Addiction; K = Korean Scale for Internet Addiction; S = Korean Smartphone Addiction scale; YIAT = Young’s Internet Addiction Test; O_A = Internet Addiction Proneness Scale for Adolescents; BDI-II = Beck Depression Inventory-II; STAI_X1 = State-Trait Anxiety Inventory X-1; RSES = Rosenberg Self-Esteem Scale; BIS = Barratt Impulsiveness Scale-II; AQ = Aggression Questionnaire; DHQ = Daily Hassles Questionnaire; JTCI_P = Junior Temperament and Character Inventory_Persistence; SD = Self-Directedness; C = Cooperativeness.
Comment 7: The data has not been interpreted properly. The third group „plain addicted” is functioning better or at least on the same level as the second group.
>>> Response 7: As you commented, we revised the group classification and the results were reinterpreted according to the revised contents.
→ The participants were divided into three subgroups (mild risk, moderate risk, addicted) according to their total DIA score. Comparisons of the clinical variables (internalizing/externalizing problems and temperament and character traits) among the DIA subgroups are shown in Tables 4. We found significant differences in Internet and smartphone addiction, depression, anxiety, self-esteem, impulsivity, aggression, and stress among the subgroups on all of the scales except the SAS-SV. The moderate risk and addicted group had significantly higher levels of Internet and smartphone addiction, anxiety, depression, impulsivity, aggression, and stress and lower self-esteem compared with the mild risk group. Moreover, scores on the JTCI, which measures temperament and character traits, revealed that the mild risk group had significantly higher levels of persistence, self-directedness, and cooperativeness than the addicted group did. (results, page 8, line 258-269)
Comment 8: There is a lack of explicit recommendation what level should be claimed as critical. If it is 3+ serious symptoms, it qualifies 81,6% of all participants. All the more, the clinical usefulness of this method is doubtful and the problem of no ROC curves comes back.
>>> Response 8: We had tried to clarify the discussion, as you have mentioned above, by reducing the number of subgroups and referring to the study process in more detail. Also, in order to perform the ROC curve analysis, it is necessary to IGD diagnosis. However, since the diagnosis of IGD is made through the DIA in this study, using it to present the cutoff of the DIA has a limitation that the DIA variable is used in duplicate. Thus, in this study, we tried to only examine the availability of grouping in the clinical practice broadly.
Comment 9: In methods it is not said that the post hoc analysis has been applied, although according to the table – it has.
>>> Response 9: Thank you for your comments. We added the post hoc analysis in the Methods.
→ “including post hoc test (Bonferronni method)” (added in the method, page 6, line 207)
Comment 10: The summary also arose some objections but its content should be fitted to corrected version of the paper.
>>> Response 10: We modified the content throughout the manuscript based on the revised version as you commented. We used track-changes function in the revised manuscript.
Comment 11: The paper has many faults and it needs general improvement. In relations to above mentioned comments I suggest: title change, redefining of aims, more precise description of procedure as well as the method of analysis, introducing the criteria of group division or revision of this division. Some of above comments can be taken into account in description of the study limitation.
>>> Response 11: We appreciated your comments. We added limitations and revised the manuscript as you commented above.
→” First, our subjects were screened using an Internet/smartphone addiction scale. Thus, it may be difficult to generalize our findings to non-clinical populations because our study did not include a control group. In addition, the proportion of the addicted group is about 80% or more, because the internet/games/SNS high-risk users were recruited in this study. Therefore, additional analysis (e.g., factor analysis, ROC curve etc.) should be made to propose a cut-off score of DIA or to make it more useful in a clinical setting.” (in the conclusion, page10, line 348-354)
Reviewer 2 Report
This a topic of clinical interest and the study is also relevant in ensuring that future clinical interventions are methodologically sound when being tailor-made across different cultures and countries. However, it is not without limitations. In order for this manuscript to be in a publishable format, it needs some significant revisions. The following shortcomings need to be addressed. Overall, the manuscript is very confusing. Throughout the paper, I was unable to ascertain if you were addressing IGD as stated in the title. The title itself seemed obscure after reading the study from start to end. The authors have stated that they assessed the clinical usefulness of DIA classifications for IGD yet that is not addressed explicitly in the study aims, and then measures for internet and smartphone addictions are used. Hence the sheer confusion. I assumed that IGD was assessed as a standalone subsection in the DIA. Parts of the puzzle only became sensical when I read the statistical analysis where it is somewhat clearer that IGD in the DIA was compared with the other measures. Other specificities are addressed below: Line 26: It would be useful to have a sentence on the background of IGD rather than going straight into the aims of the research. Line 35 and 39: Should read severely addicted. Please revise Line 52: Change "have" to "has". Line 72: changed not included to without including. Line 98: The aims are obscure as they do not match the title or the abstract. I am confused if you are referring to IGD as the subgroup or if it is indeed something completely different. Please revise. Line 109: Please make explicit your reasons for excluding children from the study when data for this group were collected. Line 151: Cronbachs alpha not reported for BDI-II Line 153: Not explicit if RSES was translated, if so, what are the psychometric properties? Table 3 and 4 need to be simplified - they are too complex and should only report the significant results. The authors have outlined the limitations of not using more IGD specific measures etc, but it appears that no specific measure for IGD was used. Rather, it was the DIA, which is why there is confusion to the reader.
Author Response
We highly appreciate your critical review that could be helpful to improve our paper. As you commented, we realized that our previous version of the manuscript had shortcomings and we tried to elaborate the manuscript throughout the paper. Detailed contents revised are as below (also marked in the revised manuscript using track-change function in MS Word).
Reviewer #2
This a topic of clinical interest and the study is also relevant in ensuring that future clinical interventions are methodologically sound when being tailor-made across different cultures and countries. However, it is not without limitations. In order for this manuscript to be in a publishable format, it needs some significant revisions. The following shortcomings need to be addressed.
Comment 1: Overall, the manuscript is very confusing. Throughout the paper, I was unable to ascertain if you were addressing IGD as stated in the title. The title itself seemed obscure after reading the study from start to end. The authors have stated that they assessed the clinical usefulness of DIA classifications for IGD yet that is not addressed explicitly in the study aims, and then measures for internet and smartphone addictions are used. Hence the sheer confusion. I assumed that IGD was assessed as a standalone subsection in the DIA.
Parts of the puzzle only became sensical when I read the statistical analysis where it is somewhat clearer that IGD in the DIA was compared with the other measures.
>>> Response 1: We appreciated your comments. We tried to make it clearer by revising the contents, title, and aim of this study throughout the manuscripts.
→ ”Application of Diagnostic Interview for Internet Addiction (DIA) in clinical practice for Korean adolescents” (modified the title)
→ “We examined the psychometric properties of the Diagnostic Interview for Internet Addiction (DIA), a semi-structured diagnostic interview tool for IGD, and verified the application of DIA in clinical practice for Korean adolescents” (in the abstract)
Other specificities are addressed below:
Comment 2: Line 26: It would be useful to have a sentence on the background of IGD rather than going straight into the aims of the research.
>>> Response 2: Thank you for your comments. We added the background of IGD in the abstract.
→ “The increased prevalence of Internet Gaming Disorder(IGD) and the inclusion of IGD in DSM-5 and ICD-11 emphasizes the importance of measuring and describing the IGD symptoms.” (added in the abstract, page 1, line 28-29)
Comment 3: Line 35 and 39: Should read severely addicted.
>>> Response 3: Thank you for comment. However, we have tried to use the term as a ‘severe addicted’ in order to maintain consistency with previous studies.
Comment 4: Please revise Line 52: Change "have" to "has".
>>> Response 4: As you commented, we changed 'have’ to 'has' (line 54).
Comment 5: Line 72: changed not included to without including.
>>> Response 5: As you commented, we changed 'not included’ to 'without including' (line 74).
Comment 6: Line 98: The aims are obscure as they do not match the title or the abstract. I am confused if you are referring to IGD as the subgroup or if it is indeed something completely different. Please revise.
>>> Response 6: We appreciated your comments. We revised the title and aim of this study to be consistent with this manuscript. And, we tried to classify the DIA subgroup by severity, which has a similar meaning to the severity of the IGD.
Comment 7: Line 109: Please make explicit your reasons for excluding children from the study when data for this group were collected.
>>> Response 7: Thank you for your comments. We added the explanation about the reasons for excluding children from this study.
→ “the sample size of children was small and there were differences between the questionnaires in children and adolescents (e.g., BDI vs. CDI, etc.)” (in the method, page 3, line 129-130)
Comment 8: Line 151: Cronbachs alpha not reported for BDI-II, Line 153: Not explicit if RSES was translated, if so, what are the psychometric properties?
>>> Response 8: As you commented, we added the cronbach’s alpha of BDI-II. Also, all the questionnaires including RSES used in this study were validated or translated in Korean. Thus, we added it in the manuscript to avoid confusion for readers.
→ “and the Cronbach’s alpha was 0.56 in this study” (in the method, page 6, line 189-190)
→ “The Rosenberg Self-Esteem Scale (RSES) [42], which is also translated in Korean” (in the method, page 6, line 192)
Comment 9: Table 3 and 4 need to be simplified - they are too complex and should only report the significant results. The authors have outlined the limitations of not using more IGD specific measures etc, but it appears that no specific measure for IGD was used. Rather, it was the DIA, which is why there is confusion to the reader.
>>> Response 9: Thank you for your comments. We tried to simplify the results by integrating Table 3 and 4 and reporting only significant results.
Round 2
Reviewer 1 Report
The authors responded to both reviews, which were very accurate and gave specific guidelines how to improve the paper. The revision significantly improved the readability of this text. I do not see any contraindications to publish the paper in this form.